# Comparative Study of Dietary Patterns by Living Arrangements: The Korea National Health and Nutrition Examination Survey (KNHANES) 2013–2015

**DOI:** 10.3390/ijerph17072371

**Published:** 2020-03-31

**Authors:** Namhee Kim, Go-Un Kim, Heejung Kim

**Affiliations:** 1College of Nursing, Yonsei University, Seoul 03722, Korea; namheekim0316@gmail.com (N.K.); kgudfc@naver.com (G.-U.K.); 2Mo-Im Kim Nursing Research Institute, Yonsei University, Seoul 03722, Korea

**Keywords:** diet, living arrangements, life style, nutrition surveys

## Abstract

This study aimed to identify the dietary patterns of Koreans, comparing them according to their living arrangements, and to determine factors associated with the patterns. We analyzed nutritional data of 6719 Korean adults aged 19–64 years from the 2013–2015 Korea National Health and Nutrition Examination Survey using the exploratory factor and hierarchical and k-means cluster analyses. We used multinominal logistic regression to compare factors associated with each dietary pattern. We identified three dietary patterns based on meal preference and dessert type: “traditional meal with healthy dessert”, “meal only”, and “unhealthy dessert” (Kaiser–Meyer–Olkin = 0.90, Bartlett’s test of sphericity *p* < 0.001). The “unhealthy dessert” dietary pattern was more frequent in people living alone (51.7%) than in those living with others (41.8%). Weight control, dining out, eating breakfast, and sleep were significantly associated with the “meal only”; eating breakfast was associated with the “unhealthy dessert” dietary pattern among those living alone. Lifestyle factors were associated with unhealthy dietary patterns in Korean adults living alone, warranting the need for a healthy diet and promotion of modifiable health behaviors for this subgroup. Thus, clinicians must provide comprehensive assessments and multidimensional interventions while considering lifestyle factors and unhealthy dietary patterns to improve the health status of them.

## 1. Introduction

Living alone is an emerging phenomenon worldwide in both developed and developing countries. Additionally, the increase in the number of people living alone is not a temporary phenomenon but an inevitable change of social structure [1]. In 2015, approximately 300 million individuals lived alone worldwide in two billion households [1]. In general, two to three tenths of the total population live alone in Europe, North America, or Asia [1]. The number of people living alone is expected to increase more rapidly by 2035 [2]. Healthcare providers and policymakers have raised concerns about this population trend as the status of living alone influences diverse health-related behaviors, such as unhealthy diet, and related consequences. Particularly, since nutrition intake and nutritional status are factors that influence health outcomes, it is necessary to examine the diet of people living alone.

Individuals living alone have different diets and relevant outcomes compared with those living with others. People living alone reportedly consume foods of lower nutritional quality, are less satisfied with their daily diet, and show more irregular dietary habits compared with those living with others [3]. A recent study reported that 33.5% of people living alone did not eat breakfast, 32.7% prepared half-cooked food, and 31.2% had food delivered [4]. Notably, people living alone have a higher rate of eating alone than those living with others, thus increasing malnutrition and obesity [5]. Such health problems are expected to increase the nation’s medical expenses [4,5]. Thus, healthcare programs and public services should be redesigned to take into consideration the conditions and the unique healthcare dietary needs of individuals living alone, as well as those living with others.

Poor dietary quality negatively affects the overall health status and is significantly associated with the prevalence of chronic diseases [6,7], it is also strongly associated with health-related behaviors [8]. Unhealthy or insufficient nutritional intake increases the chance of developing obesity, hypertension, and cardiovascular disease [6], as well as associated risk factors for non-communicable chronic diseases such as cardiovascular disorders, cancer, chronic respiratory diseases, and diabetes mellitus [7]. Unhealthy dietary patterns are also reported to be strongly associated with smoking, high alcohol consumption, and low physical activity [8]. Therefore, it is important to identify dietary patterns and examine associated factors to suggest appropriate dietary recommendations, provide tailored interventions, reduce chronic diseases, and improve the health of people living alone.

Unlike the traditional nutritional epidemiologic methods focusing on nutrients or foods of individuals, dietary patterns that focus on the complexity of the overall diet and represent broader nutrient consumption have been widely used [9]. Both the effect of an overall diet on chronic diseases and its correlation with environmental factors and risk of disease can be confirmed by identifying dietary patterns through factor and cluster analyses, using data derived from the food frequency questionnaire (FFQ) [9]. The FFQ gathers qualitative information regarding the frequency of food items that have been consumed in the past 12 months. Moreover, it is commonly used to identify dietary patterns related to clusters of the frequency of food consumption [9,10]. The FFQ has been used as a reasonable tool to identify the associations between dietary patterns and health behaviors [8], risk factors of diseases [6], and trends in specific populations [11].

Very limited reports are available on the dietary patterns of individuals living alone in Korea [12]. Few studies have investigated the general factors that are reported to be associated with diet [12,13], such as culture [12] and generational difference [12,13]. No study to date has looked into the association of dietary patterns and living arrangements, specifically on the status of living alone in Korea. Moreover, most previous studies have focused on the association of dietary patterns with non-modifiable characteristics, such as advanced age or low income in small or specific groups [4,8,13]. In this study, we aimed to identify general dietary patterns of Korean adults using national data and compare the dietary patterns of individuals living alone and those living with others. Additionally, we examined unique factors associated with the dietary patterns of the two groups.

## 2. Materials and Methods

### 2.1. Study Design

This is a cross-sectional and descriptive correlational study using secondary data analysis. 

### 2.2. Data Source and Study Sample

Primary data were obtained from the Korea National Health and Nutrition Examination Survey (KNHANES) VI conducted by the Korea Centers for Disease Control and Prevention. The data were collected from January 2013 to December 2015 and released in January 2017 [10]. KNHANES is a nationwide survey to examine the health and nutritional status of non-institutionalized individuals in the Republic of Korea. It consists of a health interview, health examination, and nutrition survey of all household members.

A stratified, multistage, and clustered probability sampling design depending on region size and demographic characteristics was used to collect preliminary data. In KNHANES VI, the final sampling was conducted in 576 regions and included 11,520 households, covering 22,948 individuals [10]. Participants were asked to answer a structured questionnaire (response rate = 78.3%). After the health interview and examination survey, the nutrition survey was completed by dietitians visiting 6719 adults aged 19–64 years in their homes [10], which excluded elderly ≥65 years. Using secondary data analysis, we identified two subgroups according to living arrangement: 6318 individuals who lived with others and 401 who lived alone in young and middle-aged groups.

### 2.3. Measurements

#### 2.3.1. The Food Frequency Questionnaire

The FFQ was a self-reported questionnaire to evaluate young and middle-aged adults’ diet and not for the elderly. The nutrition survey consisted of self-reports on dietary survey, food frequency, food intake, and food stability to assess the frequency of consuming any of the 112 food items listed there through asking the question “How often did you consume each food during the last year?”. There were 9 levels of response, 1 = rarely; 2 = once per month; 3 = twice or three times per month; 4 = once per week; 5 = twice or up to four times per week; 6 = five or six times per week; 7 = once per day; 8 = twice per day; and 9 = three times per day [10]. These food items were classified into 22 food groups based on their nutrients profile, raw material content, culinary requirement, or moisture content [10], items used in other studies [11] and based on well-established in psychometrics [14].

#### 2.3.2. Study Variables

Independent variables were categorized according to sociodemographic and lifestyle characteristics. Sociodemographic characteristics included age, sex, marital status, economic status, education, and living arrangement. Lifestyle characteristics included smoking, drinking, exercise, weight control, dining out, eating breakfast, and sleep [10]. Living arrangements were categorized into two groups: living with others (coded as 0; households of two people or more) and living alone (coded as 1; household of one) by asking the question “Which of the following is your household type?”. All variables were re-categorized as in previous studies to compare findings [10,15,16].

#### 2.3.3. Data Procedures

Two investigators obtained the publicly available data [10] and cross-checked them for completeness in terms of (a) matching the computed data with the available answers in the questionnaires; (b) checking and setting the missing values based on reasonably possible ranges; and (c) skimming through the frequencies of each study variables to assure the distribution considering outliers, especially ensuring the dietary behaviors, FFQ, and independent variables were checked [10]. For all study variables, missing data accounted for 0.4–12.7% and were reported as “non-response” or “unknown”. Among the 6719 individuals, 898 missing cases (13.4%) were identified in the multinominal logistic regression model because on the listwise deletion regarding independent variables. We conducted multiple imputation by chained equations, which estimated the value of the missing data using the distribution of observed data [17,18]. To reduce the amount of missing data for all variables tested to 2.2% [19]. The weight of the 3-year period (2013–2015) was integrated to generate a single weight value based on the followed equation [10]: 2013–2015 association analysis weight (investigation part: health interview, health examination, and nutrition survey) × (192 annually regions/576 total regions). A complex sample design was generated for the estimation of variance and proper calculation through the stratum (stratification variable), primary sampling unit (cluster variable), and weight (sample weight variable) [10]. All analyses were completed using the weighted data of the complex sample design with the add-on module of IBM SPSS Complex Samples (IBM Co., Armonk, NY, USA).

### 2.4. Data Analysis

All statistical analyses were conducted using IBM SPSS version 24 and the add-on module of IBM SPSS Complex Samples. Significance level was set at 0.05 (two-tailed). To identify dietary patterns, principal components analysis (PCA) within the exploratory factor analysis (EFA) approaches, hierarchical and k-means cluster analyses were performed. The hierarchical cluster analysis was performed using the standardized z-scores, Ward’s method, and the squared Euclidean distance measure. The k-means cluster analysis was performed using the Caliński–Harabasz stopping rules for 2–5 cluster solutions [20]. Multinominal logistic regression analysis was used to compare the factors of each identified dietary pattern in people living alone, to those living with others. We tested for possible multicollinearity among the 13 independent variables and 22 food groups based on correlation analysis and a collinearity diagnostic test. No statistical issue was observed in terms of multicollinearity, Durbin–Watson statistic, and multivariate outliers.

### 2.5. Ethical Considerations

The primary data collection was approved by the Institutional Review Board (IRB) of the Korea Centers for Disease Control and Prevention (IRB numbers 2013-07CON-03-4C and 2013-12EXP-03-5C). Written informed consent was obtained from all subjects and they participated voluntarily according to the National Health Enhancement Act. This secondary data analysis study was approved with exempt status by the IRB of the affiliated university (IRB number Y-2017-0092).

## 3. Results

### 3.1. Characteristics of Participants by Living Arrangement

Among the participants, 93.8% of lived with others and 6.2% lived alone. The characteristics of people living alone and those living with others were different. Age, sex, marital status, economic status, education, smoking, drinking, dining out, and eating breakfast were factors associated with a living arrangement and they were statistically significant (*p* < 0.05). Details on the sociodemographic and lifestyle characteristics of the participants are shown in Table 1.

### 3.2. Identification of Dietary Patterns

To identify different dietary types based on the 22 food groups, EFA was conducted with PCA. The results of Kaiser–Meyer–Olkin (0.90) and Bartlett’s test of sphericity (χ^2^ = 290821.97; df = 231; *p* < 0.001) showed a sufficient sample size and inter-item correlations. Eigenvalues of > 1 were used to identify the number of factors, while a scree plot was rotated by an orthogonal transformation using a varimax rotation to obtain a more condensed structure with greater interpretability [21]. Four factor scores were generated by factor loading using the EFA. Each dietary type was labeled as a “substitute meal (factor 1)”, “traditional Korean meal (factor 2)”, “healthy dessert (factor 3)”, or “unhealthy dessert (factor 4)”. These four factors explained 52.9% of the total variance, consisting 17.7% for “substitute meal”, 15.6% for “traditional Korean meal”, 10.4% for “healthy dessert”, and 9.2% for “unhealthy dessert” (Table 2). 

Hierarchical and k-means cluster analyses were conducted using the identified four factor scores of dietary types. We used the three-cluster solution for dietary patterns and labeled them as follows: “traditional meal with healthy dessert” (cluster 1), “meal only” (cluster 2), and “unhealthy dessert” dietary patterns (cluster 3; Table 3). Both clusters 1 and 2 were based on the “traditional Korean meal”. Cluster 1 (22.0%, the smallest cluster) was associated with a large consumption of a “healthy dessert”, cluster 2 (35.6%) with a high consumption of a “substitute meal” along with a “traditional Korean meal”, and cluster 3 (42.4%, the largest group) with more consumption of an “unhealthy dessert” but less consumption of a “substitute meal”, “traditional Korean meal”, or “healthy dessert” (Figure 1).

### 3.3. Dietary Patterns and Characteristics of Participants by Living Arrangement

Further, among people living with others, 41.8% had “unhealthy dessert”, 35.9% reported “meal only”, and 22.3% had a “traditional meal with healthy dessert” dietary pattern. Similarly, among people living alone, 51.7% had “unhealthy dessert”, 30.9%, “meal only”, and 17.4% had a “traditional meal with healthy dessert” dietary pattern; apparently, the percentage of “unhealthy dessert” consumption was higher in people living alone in contrast to those living with others (Table 4). 

People living alone and those living with others followed different dietary patterns. Age, sex, marital status, education, smoking, drinking, dining out, and eating breakfast were the statistically significant factors associated with the different dietary patterns (*p* < 0.05). Details on the dietary patterns, sociodemographic, and lifestyle characteristics of the participants by living arrangement are shown in Table 4. 

### 3.4. Factors Associated with Dietary Patterns depending on Living Arrangement

To examine the associations between lifestyle characteristics and dietary patterns adjusted for sociodemographic characteristics, we used the multinominal logistic regression analyses (Table 5), with the dietary pattern of the “traditional meal with healthy dessert” (cluster 1) chosen as the reference. The regression models with sociodemographic and lifestyle factors significantly explained the dietary patterns regardless of living arrangement (Cox and Snell R^2^ = 0.29 and Nagelkerke R^2^ = 0.33 in people living with others; Cox and Snell R^2^ = 0.43 and Nagelkerke R^2^ = 0.50 in people living alone). Although most independent variables were significantly associated with dietary patterns in the group who lived with others, there were different associations of dietary patterns with age, marital status, weight control, dining out, eating breakfast, and sleep factors in the group who lived alone.

Moreover, among people living with others, most variables were significantly related to a “meal only” (cluster 2) or “unhealthy dessert” dietary pattern (cluster 3) compared with a “traditional meal with healthy dessert” dietary pattern (cluster 1). Men or those with a poor lifestyle (smoking, alcohol drinking, lack of exercise, or lack of effort for weight control) had a “meal only” or “unhealthy dessert” dietary pattern. Advanced age, being married, and high economic status were associated with a higher tendency of having a “traditional meal with healthy dessert” than a “meal only” or “unhealthy dessert” dietary pattern. In addition, the “meal only” dietary pattern was associated with 6–8 h of sleep per day and dining out less than once a week. Low education level and eating fewer breakfast were associated with a “meal only” dietary pattern (elementary school or less: odds ratio (OR) = 0.65, 95% confidence interval (CI) 0.43 – 0.98; eating no or less frequent breakfast: ranges of ORs from 1.37 to 1.83, *p* < 0.05); or an “unhealthy dessert” dietary pattern (elementary school or less: OR = 1.73, 95% CI 1.32–2.27; eating no or less frequent breakfast: ranges of ORs from 1.48 to 2.55, *p* < 0.01; Table 5).

Furthermore, among people living alone, two variables were found to be equally significant: (a) older age was related to a “traditional meal with healthy dessert” dietary pattern and (b) 6–8 h of sleep per day and dining out less frequently were related to a “meal only” dietary pattern. However, several differences were observed. First, marital status, such as being separated, widowed, or divorced, was associated with a “meal only” dietary pattern (OR = 0.23, 95% CI 0.07–0.77). Second, no effort for weight control among the health-promoting behaviors was also associated with a “meal only” dietary pattern (OR = 3.13, 95% CI 1.14–8.54). Third, having no or 3–4 time breakfast was associated with a “meal only” (no: OR = 3.77, 95% CI 1.39–10.22; 3-4 times/week: OR = 6.59, 95% CI 1.74–27.81) or “unhealthy dessert” (no: OR = 4.45, 95% CI 1.80–11.02; 3-4 times/week: OR = 3.97, 95% CI 1.24–12.76) dietary pattern; however, the associations were stronger in people living alone than in those living with others.

## 4. Discussion

This study identified three major dietary patterns, namely “traditional meal with healthy dessert”, “meal only”, and “unhealthy dessert”, in young and middle-aged Korean adults. People living alone had an “unhealthy dessert” dietary pattern more often than those living with others. Dietary patterns showed significant associations with lifestyle factors. The “meal only” dietary pattern of people living with others was associated with all lifestyle factors, while the “unhealthy dessert” dietary pattern was affected by all lifestyle factors except for dining out and sleep. Particularly, in people living alone, the “meal only” dietary pattern was strongly affected by lifestyle factors, such as no previous effort of weight control, 1–3 times dining out, no or some irregular eating breakfast, and shorter sleep, while an “unhealthy dessert” dietary pattern was affected by no or some irregular eating breakfast. 

The three dietary patterns have similarities with those of previous studies as well as differences. These patterns were identified by combining two meals and two dessert dietary types: “substitute meal”, “traditional Korean meal”, “healthy dessert”, and “unhealthy dessert” dietary types. A previous study, using the 1998–2005 KNHANES data, reported that the traditional Korean food pattern mainly involves consumption of raw vegetables, fish, kimchi, and seaweed, while the modified pattern mainly involves consumption of bread, noodles, and fast foods [11]. Moreover, the dietary patterns of Korean adults aged > 20 years comprised of a “Korean diet” (kimchi, vegetables, and rice), a “Western diet” (eggs, oil, and soda), and a “new diet” (fruits, dairy products, and potatoes) from 1998 to 2010 [22]. Our findings demonstrated similar patterns with regard to the “traditional meal with healthy dessert” and the “meal only” dietary patterns. However, the “unhealthy dessert” dietary pattern was a newly identified pattern in this study. “Unhealthy dessert” was mainly sugar-sweetened desserts including cookies, chocolates, ice creams, sugar, jam, coffee, and carbonated beverages. There has been a societal transition from meal-centered dietary patterns to consumption of substitute meals and desserts in Korea in accordance with changes in lifestyle [11].

Koreans are less likely to have a “traditional meal with healthy dessert” dietary pattern and more likely to have either a “meal only” or an “unhealthy dessert” dietary pattern. Specifically, people living alone are more likely to have an “unhealthy dessert” dietary pattern compared to those living with others. Additionally, there are vulnerable groups of individuals who manage their own health and who have insufficient healthy behaviors and have various health problems [5,13]. Moreover, unhealthy diets are significantly associated with health problems, thus attention should be paid to such patterns to prevent related health issues [5,6,7,8]. According to a systematic review of the relationship between living alone and food and nutrient intake [23], people living alone were found to be less likely to follow a varied diet and have lower fruit, vegetable, and fish consumption than those living with others. Similarly, this study showed that the tendency to have a “traditional meal with healthy dessert” dietary pattern including vegetables, beans, fish, soup, seaweed, stew, hard plants, fruits, healthy drinks, and nuts was lower in people living alone than in those living with others. Furthermore, half of the people living alone followed an “unhealthy dessert” dietary pattern including saturated fats, beverages, and sugar-based desserts. Particularly, sugar-sweetened beverages are associated with a high prevalence of type 2 diabetes, cardiovascular disease, osteoporosis, and dental caries [24] and it is known to increase mortality by 26.8% over a 10-year period [25]. While, healthy foods help prevent obesity, hypertension, high cholesterol, and hyperglycemia [25]. Thus, it is important to assess the unhealthy dietary pattern including the level of unhealthy dessert consumption in young and middle-aged adults of people living alone.

Secondly, Korean adults are likely to have a “meal only” dietary pattern, whether they live alone or not. In this study, the “meal only” dietary pattern consisted of both “substitute” and “traditional Korean meals”. According to studies that examined the change in dietary patterns from 1998 to 2005, the dietary pattern of Koreans changed from the traditional pattern, i.e., consumption of rice-based meals, to the modified pattern, i.e., consumption of meat and processed foods with noodles and bread as the main meal ingredients [11]. Additionally, this study found that the “meal only” dietary pattern also included fast foods, meat, processed fish products, grains, noodles, and bread. More than half of the Koreans living alone consumed ready-to-eat foods, which were the main components of the “meal only” dietary pattern [13]. It is hypothesized that the demand for simple food and delivered food that can be easily prepared at home is influenced by the unique eating habits of people living alone [3]. People living alone purchase a small amount of food at convenience stores and via mail-order sales, in contrast to meal preparation at home by those living with others [3]. Foods such as fast foods, meats, and flour, which can be consumed easily and quickly, can increase blood sugar and fracture risk [26]. Therefore, it is necessary to pay attention to nutritional imbalances and unique eating habits to develop appropriate dietary interventions for people living alone.

In this study, the identified dietary patterns demonstrated significant associations with lifestyle factors. Individuals who smoke, do not exercise, and have no weight control concerns are less likely to have a healthy dietary pattern [27]. In this study, individuals living with others who consumed alcohol reported having “meal only” and “unhealthy dessert” dietary patterns. Smoking, alcohol consumption, and physical inactivity are generally considered as unhealthy behaviors [8,28]. These unhealthy behaviors and poor dietary patterns are known to be closely correlated [28]. According to the global burden of diseases, the death rate due to poor dietary patterns has increased by 11%, and there has been a 4% increase in smoking-related mortality in the last 10 years [25]. This combination of unhealthy behaviors and dietary patterns is expected to result in high mortality and morbidity. Thus, the complex nature of multiple health behaviors and dietary patterns must be considered when developing interventions and public programs to promote a healthy dietary pattern.

Regarding the living-alone group, weight control, dining out, eating breakfast, and sleep were significantly associated with the “meal only” dietary pattern. The “meal only” dietary pattern of people living alone was strongly affected by lifestyle factors such as weight control concerns and sleep, which is consistent with the findings in previous studies [27,29]. Leech et al. [30] observed that individuals did not sufficiently consume healthy foods such as vegetables, fish, fruits, and nuts when they did not exercise. Furthermore, a healthy diet is an important factor for improving the duration and quality of sleep [29]. In this study, lifestyle factors such as weight control and sleep were significantly associated with people living with others and living alone; however, the association was greater in those living alone. Therefore, this should be considered when developing or conducting nutritional education programs and policies.

Generally, Koreans living alone tend to eat out or have an irregular mealtime and lifestyle [3,13]. According to the 2016 Korean food consumption pattern survey data, people living alone had fewer meals at home [31]. Additionally, they consumed simple meals more frequently or used ready-to-eat meals purchased from convenience stores, street food facilities, or restaurants contrary to people living with others [31]. Meanwhile, consumption of convenience foods such as processed foods, delivery foods, and fast foods [12] has increased due to insufficient time for preparing meals. The “meal only” and “unhealthy dessert” dietary patterns observed in our study are similar to the use of ultra-processed foods, which largely comprise ready-to-eat foods and are the major causes of non-communicable diseases [32]. Therefore, it is important to improve dietary patterns that include the consumption of convenience foods, such as ultra-processed foods, it is likely to be important to maintain the health of Koreans living alone. Furthermore, it is necessary to educate people who live alone regarding healthy diet choices when dining out, reflecting the characteristics of the living environment of someone who lives alone, and eats out.

Eating no breakfast or 1–4 times a week was associated with unhealthy dietary patterns. Particularly, the association between eating breakfast and dietary patterns was evident in people living alone than those living with others. The proportion of Koreans living alone who consumed breakfast less than once a week was 34.6% in 2012 [12]. Similarly, the percentage of people living alone who did not consume breakfast was 34.1% in our study. Non-regular consumption of breakfast is highly associated with an unhealthy diet [27]. According to the meal pattern and nutrient intake and quality literature review, breakfast-skipping and diet quality were inversely associated, while skipping breakfast was related to lower micronutrient intake and resulted in chronic illnesses [30]. Furthermore, according to the same systematic review, breakfast-skipping was associated with overweight and obesity and increased their risks [33]. Skipping breakfast led to improper eating habits such as irregular meals, not eating slowly, and not eating a variety of foods [34]. Therefore, nutrition education and interventions are needed to prevent nutritional deficiencies caused by skipping breakfast and to ensure proper eating habits and behaviors.

### 4.1. Practical Implications

To date, there is a lack of research on diet among people living alone in the community, while related dietary policies are insufficient compared to those in residential or medical care. Our study findings should be carefully considered for concluding practical implication because of missing information of FFQ among elderly. Current statistics of Korean adults living alone were reported as 29.2% when including elderly [35]. Our study samples had much smaller size of those living alone, 6.2% in weighted samples, because the primary data were collected with only young and middle-aged adults. Thus, practical implication should be developed for young and middle-aged adults rather than elderly.

Nonetheless, the advantages of this study pertained to the identification of the dietary patterns in relation to living arrangements as well as to raising awareness regarding the relationship between dietary patterns and lifestyle factors. Particularly, people living alone had significantly higher unhealthy dietary pattern and were more likely to be affected by lifestyle factors. This study also provided information regarding intervention and policy development for individuals, groups, and populations. As discussed, healthcare providers should consider the weight control and sleep characteristics of individuals who live alone and additionally encourage regular breakfast intake to promote a balanced diet when dining out. In 2016, Korea enacted the “framework ordinance for people living alone support” and was implemented in Seoul, Korea’s capital, in 2019. Considering the growth of the population living alone, this program is expected to expand to suburban and urban areas gradually by 2023 [36]. The plan was to establish a communal kitchen by linking cooking schools and cultural centers in the community, as well as to implement a “social dining program” where people living alone gather to prepare food, eat, and communicate [36]. The community-based program should be tailored to the nutrition strategies for people living alone. It should focus on improving cooking skills and sharing recipes and ideas regarding meal preparation, including food purchase, preparation, and storage methods to improve the availability of healthy foods [23]. Additionally, budgets must be secured so that the developed policies can be well-implemented, and a legal basis must be introduced to establish standards. Furthermore, a task force linked to government agencies and private organizations must be established to promote diet and health for individuals living alone.

### 4.2. Limitations

Our study has several limitations. First, we used cross-sectional data, limiting our ability to infer causality. Second, we did not include adults aged ≥65 years. Although Koreans who live alone generally belong to the aged group, our KNHANES-based nutrition survey was performed only in adults aged 19–64 years. Third, we did not use variables that reflected the characteristics of living alone or dietary patterns such as eating behavior and regularity of eating habits because of limited information on variables from the secondary data analysis. Finally, our study has a wide disparity in the sizes of samples, living with other vs. living alone, approximately 15.1:1, which make us difficult to compare factors between two groups. Further studies should be conducted using propensity score matching, reflecting our identified covariates to overcome possible bias due to unbalanced sample size. In addition, future prospective and longitudinal studies focusing on extensive data collection and considering the aging subgroups, current disease, and health condition are recommended. We also suggest that further primary data collection with comprehensive assessments and multidimensional interventions to promote healthier diets in people living alone and those living with others are needed.

## 5. Conclusions

Our study identified three dietary patterns (“traditional meal with healthy dessert”, “meal only”, and “unhealthy dessert”) in Korean young and middle-aged adults using representative data from the national sample. Particularly, “unhealthy dessert” dietary pattern was higher in people living alone than in those living with others. We also found that factors related to dietary patterns differed according to their living arrangement. The dietary patterns of Korean adults were associated with sociodemographic characteristics, and particularly lifestyle characteristics such as weight control concerns, dining out, eating breakfast, and sleep. Particularly, lifestyle factors were associated with unhealthy dietary patterns in Korean adults living alone. These findings could be fundamental for developing nutritional education programs and policies specifically targeting individuals who are living alone. Furthermore, the results could improve the promotion of self-care and decrease the incidence of various diseases caused by poor diet by advocating proper eating habits for people living alone.

## Figures and Tables

**Figure 1 ijerph-17-02371-f001:**
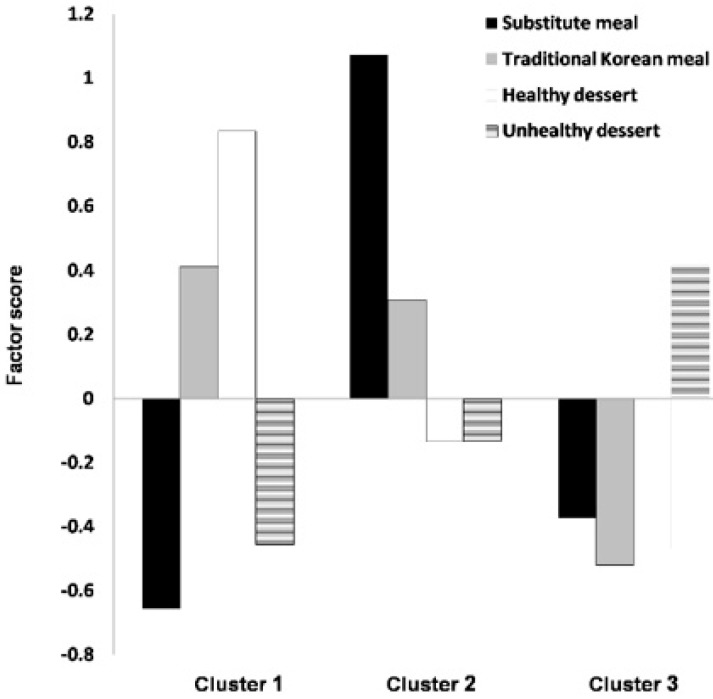
Mean values of centered cluster subjects.

**Table 1 ijerph-17-02371-t001:** Characteristics of participants by living arrangement (*n* = 6719).

Variables	Categories	Total (*n* = 6719)	Living with Others (*n* = 6318)	Living Alone (*n* = 401)	χ^2^	*p*
		100.0%	93.8%	6.2%		
Age	19–34 years	34.6	34.0	43.2	210.67	0.001
	35–49 years	36.0	37.0	22.2		
	50–64 years	29.4	29.0	34.6		
Sex	Men	50.2	49.5	61.6	130.55	0.001
	Women	49.8	50.5	38.4		
Marital status	Single	27.5	25.1	63.1	4114.93	<0.001
	Married	66.8	70.4	10.9		
	Separated, widowed, or divorced	5.7	4.5	26.0		
Economic status	Lowest	8.4	7.5	22.0	671.70	<0.001
	Lower-middle or upper-middle	56.3	56.3	56.0		
	Highest	35.3	36.2	22.0		
Education	Elementary school or less	6.9	6.3	14.2	208.98	<0.001
	Middle and high school	48.7	49.0	44.9		
	Some college or higher	44.4	44.7	40.9		
Smoking	Past smokers or non-smokers	76.6	77.4	64.3	213.52	<0.001
	Current smokers	23.4	22.6	35.7		
Drinking	Non-alcohol drinking	38.0	38.4	31.1	49.76	0.008
	Alcohol drinking	62.0	61.6	68.9		
Exercise	Less than one day a week or never	77.6	77.7	76.2	2.73	0.538
	More than two days a week	22.4	22.3	23.8		
Weight control	Never tried to control the weight	29.3	29.3	29.3	0.01	0.978
	Effort	70.7	70.7	70.7		
Dining out	Less than once a month	2.7	2.7	3.1	243.30	<0.001
	1–3 times a month	13.8	13.8	12.9		
	1–6 times a week	50.8	51.4	41.2		
	Once a day	21.1	21.1	21.7		
	More than twice a day	11.6	11.0	21.1		
Eating breakfast	None a week	17.1	16.0	34.1	531.40	<0.001
	1–2 times a week	12.7	12.7	13.0		
	3–4 times a week	15.1	15.4	11.0		
	5–7 times a week	55.1	55.9	41.9		
Sleep	0–5 h a day	12.7	12.5	16.0		
	6–8 h a day	80.9	81.0	78.7	27.59	0.169
	9 h or more a day	6.4	6.5	5.3		

Note. The missing data was less than 5% and non-responses were excluded from the analysis. All percentages presented here were weighted using a complex sample design.

**Table 2 ijerph-17-02371-t002:** Factor loading matrix for the four major dietary types (*n* = 6719).

Food Groups	Factor 1	Factor 2	Factor 3	Factor 4
	Substitute Meal	Traditional Korean Meal	Healthy Dessert	Unhealthy Dessert
Fast foods	**0.81**	−0.08	−0.06	0.15
Meat	**0.70**	0.33	0.01	0.07
Processed fish products	**0.70**	0.15	−0.01	0.11
Poultry	**0.67**	0.23	−0.01	0.03
Grains	**0.55**	0.19	0.12	0.07
Noodles	**0.54**	0.24	0.09	0.08
Bread	**0.49**	−0.11	0.45	0.23
Rice cakes	**0.47**	0.03	0.45	−0.03
Eggs	**0.43**	0.29	0.22	−0.05
Vegetables	0.18	**0.78**	0.29	0.05
Beans	0.09	**0.71**	0.19	0.00
Fish	0.13	**0.70**	0.22	0.06
Soup	0.31	**0.64**	0.11	0.07
Seaweeds	0.10	**0.61**	0.19	0.06
Stew	0.36	**0.60**	−0.04	0.07
Hardy plants	−0.08	0.23	**0.71**	−0.08
Fruits	0.07	0.30	**0.68**	−0.06
Healthy drinks	0.27	0.14	**0.54**	0.08
Nuts	−0.14	0.34	**0.52**	0.02
Saturated fats	−0.08	0.07	−0.07	**0.87**
Beverages	0.17	0.14	−0.07	**0.75**
Sugar-based desserts	0.39	−0.03	0.20	**0.75**
Total % of variance explained	17.7%	15.6%	10.4%	9.2%

Note. Numbers in bold indicate that factor loading is more than the cut-off score of 0.40 and it belongs to each column.

**Table 3 ijerph-17-02371-t003:** Results of cluster analysis using four factor scores (*n* = 6719).

Dietary Types	Cluster 1 (22.0%)	Cluster 2 (35.6%)	Cluster 3 (42.4%)	*F*	*p*
Substitute meal	−0.66	1.07	−0.37	23,473.10	<0.001
Traditional Korean meal	0.41	0.31	−0.52	4651.37	<0.001
Healthy dessert	0.84	−0.13	−0.47	8245.33	<0.001
Unhealthy dessert	−0.45	−0.13	0.41	3098.75	<0.001

Note. Cluster 1, “traditional meal with healthy dessert” dietary pattern; cluster 2, “meal only” dietary pattern; cluster 3, “unhealthy dessert” dietary pattern. All percentages presented here were weighted using a complex sample design.

**Table 4 ijerph-17-02371-t004:** Dietary patterns and characteristics of participants by living arrangement (*n* = 6719).

Living Arrangement	Living with Others (*n* = 6318)					Living Alone (*n* = 401)				
Variables/Categories	Traditional Meal with Healthy Dessert (*n* = 1768)	Meal Only (*n* = 2018)	Unhealthy Dessert (*n* = 2532)	χ^2^	*p*	Traditional Meal with Healthy Dessert (*n* = 107)	Meal Only (*n* = 87)	Unhealthy Dessert (*n* = 207)	χ^2^	*p*
Total Weighted %	22.3%	35.9%	41.8%			17.4%	30.9%	51.7%		
Age										
19–34 years	12.4	49.2	32.6	6306.88	<0.001	4.5	73.3	38.1	650.73	<0.001
35–49 years	28.2	40.6	38.5			21.3	17.5	25.3		
50–64 years	59.4	10.2	28.9			74.2	9.2	36.6		
Sex										
Men	26.8	54.9	56.9	2133.62	<0.001	37.3	64.9	67.8	128.06	0.002
Women	73.2	45.1	43.1			62.7	35.1	32.2		
Marital status										
Single	8.7	34.0	26.3	1887.21	<0.001	19.4	89.6	61.9	576.85	<0.001
Married	85.4	64.0	67.9			25.1	5.1	9.6		
Separated, widowed, or divorced	5.9	2.0	5.8			55.5	5.3	28.5		
Economic status										
Lowest	7.6	5.0	9.8	503.95	<0.001	24.0	12.0	27.3	72.33	0.073
Lower-middle or upper-middle	51.5	55.3	59.6			53.8	60.4	54.2		
Highest	40.9	39.7	30.6			22.2	27.6	18.5		
Education										
Elementary school or less	11.1	1.2	8.3	1495.99	<0.001	24.3	2.2	17.9	186.38	<0.001
Middle and high school	52.8	43.9	51.3			49.0	41.8	45.4		
Some college or higher	36.1	54.9	40.4			26.7	56.0	36.7		
Smoking										
Past smokers or non-smokers	91.8	76.2	70.8	1320.93	<0.001	78.5	71.6	55.1	99.72	0.007
Current smokers	8.2	23.8	29.2			21.5	28.4	44.9		
Drinking										
Non-alcohol drinking	54.6	28.7	38.1	1388.85	<0.001	42.8	18.6	34.7	88.12	0.002
Alcohol drinking	45.4	71.3	61.9			57.2	81.4	65.3		
Exercise										
Less than one day a week or never	76.6	75.6	80.1	86.07	0.004	81.8	68.6	78.9	35.15	0.102
More than two days a week	23.4	24.4	19.9			18.2	31.4	21.1		
Weight control										
Never tried to control the weight	23.6	28.7	33.0	223.95	<0.001	23.1	30.2	30.8	9.34	0.550
Effort	76.4	71.3	67.0			76.9	69.8	69.2		
Dining out										
Less than once a month	4.6	0.3	3.7	2348.96	<0.001	4.7	0.0	4.4	379.71	<0.001
1–3 times a month	23.7	5.6	15.6			33.7	0.7	13.3		
1–6 times a week	51.5	54.5	48.8			45.3	41.3	39.7		
Once a day	14.2	24.8	21.5			7.4	26.9	23.3		
More than twice a day	6.0	14.8	10.4			8.9	31.1	19.3		
Eating breakfast										
None a week	8.0	15.0	21.1	1896.77	<0.001	9.9	39.9	38.7	301.99	<0.001
1–2 times a week	6.7	16.0	13.0			11.3	18.9	10.2		
3–4 times a week	10.5	20.0	14.1			4.6	16.0	10.1		
5–7 times a week	74.8	49.0	51.8			74.2	25.2	41.0		
Sleep										
0–5 h a day	15.4	9.9	13.1	194.43	<0.001	24.6	9.2	17.2	59.57	0.121
6–8 h a day	79.4	84.0	79.4			72.1	86.5	76.2		
9 h or more a day	5.2	6.1	7.5			3.3	4.3	6.6		

Note. The missing data was less than 5% and non-responses were excluded from the analysis. All percentages presented here were weighted using a complex sample design.

**Table 5 ijerph-17-02371-t005:** Multinominal logistic regression to compare factors associated with dietary patterns depending on living arrangement (*n* = 6719).

Variables/Categories	Living with Others				Living Alone			
	Meal Only Dietary Pattern		Unhealthy Dessert Dietary Pattern		Meal Only Dietary Pattern		Unhealthy Dessert Dietary Pattern	
	OR	95% CI	OR	95% CI	OR	95% CI	OR	95% CI
**Sociodemographic characteristics**								
Age (ref. 19–34 years)								
35–49 years	0.42 ***	0.30, 0.58	0.75	0.54, 1.03	0.10 **	0.02, 0.44	0.15 *	0.03, 0.65
50–64 years	0.06 ***	0.04, 0.09	0.25 ***	0.18, 0.35	0.04 **	0.01, 0.23	0.08 **	0.02, 0.42
Sex (ref. Women)								
Men	2.73 ***	2.17, 3.42	3.47 ***	2.80, 4.29	2.37	0.83, 6.79	2.45	0.99, 6.02
Marital status (ref. Single)								
Married	0.92	0.64, 1.32	0.67 *	0.47, 0.96	0.27	0.06, 1.21	0.52	0.16, 1.62
Separated, widowed, or divorced	0.72	0.42, 1.25	0.89	0.56, 1.41	0.23 *	0.07, 0.77	0.56	0.21, 1.45
Economic status (ref. Lower-middle or upper-middle)								
Lowest	1.03	0.69, 1.53	1.19	0.85, 1.67	0.81	0.27, 2.43	1.83	0.88, 3.81
Highest	0.81 *	0.67, 0.98	0.68 ***	0.57, 0.82	0.76	0.25, 2.27	0.61	0.23, 1.58
Education (ref. Middle and high school)								
Elementary school or less	0.65 *	0.43, 0.98	1.73 ***	1.32, 2.27	0.72	0.13, 4.08	1.75	0.72, 4.24
Some college or higher	1.18	0.98, 1.43	0.95	0.79, 1.13	0.65	0.26, 1.64	0.74	0.34, 1.63
**Lifestyle characteristics**								
Smoking (ref. Past smokers or non-smokers)								
Current smokers	1.53 **	1.12, 2.09	1.89 ***	1.43, 2.49	0.79	0.23, 2.66	2.01	0.74, 5.44
Drinking (ref. Non-alcohol drinking)								
Alcohol drinking	1.74 ***	1.45, 2.08	1.21 *	1.03, 1.43	1.41	0.58, 3.42	0.79	0.41, 1.53
Exercise (ref. More than two days a week)								
Less than one day a week or never	1.38 **	1.12, 1.71	1.68 ***	1.37, 2.05	0.59	0.22, 1.54	0.63	0.27, 1.45
Weight control (ref. Effort)								
Never tried to control the weight	1.21 *	1.00, 1.47	1.27 **	1.06, 1.52	3.13 *	1.14, 8.54	1.74	0.73, 4.18
Dining out (ref. More than twice a day)								
Less than once a month	0.10 ***	0.04, 0.27	1.02	0.64, 1.61	< 0.001		0.47	0.10, 2.32
1–3 times a month	0.29 ***	0.19, 0.45	0.89	0.62, 1.28	0.02 ***	0.00, 0.16	0.31	0.09, 1.08
1–6 times a week	0.73	0.51, 1.05	0.93	0.67, 1.30	0.38	0.13, 1.13	0.40	0.15, 1.09
Once a day	0.80	0.55, 1.15	0.99	0.68, 1.43	0.79	0.15, 4.26	1.13	0.23, 5.50
Eating breakfast (ref. 5–7 times a week)								
None a week	1.37 *	1.01, 1.86	2.55 ***	1.91, 3.41	3.77 **	1.39, 10.22	4.45 **	1.80, 11.02
1–2 times a week	1.83 ***	1.33, 2.51	1.95 ***	1.43, 2.66	0.64	0.16, 2.60	0.48	0.12, 1.98
3–4 times a week	1.62 **	1.22, 2.16	1.48 **	1.14, 1.92	6.95 **	1.74, 27.81	3.97 *	1.24, 12.76
Sleep (ref. 0–5 h a day)								
6–8 h a day	1.45 **	1.11, 1.89	1.13	0.90, 1.43	3.63 *	1.19, 11.12	1.51	0.69, 3.32
9 h or more a day	1.21	0.80, 1.84	1.16	0.80, 1.69	0.76	0.05, 11.68	0.87	0.08, 9.10
Cox and Snell R^2^	0.29				0.43			
Nagelkerke R^2^	0.33				0.50			

Note. Adjusted for age, sex, marital status, economic status, and education. The reference class among the dependent variables was the “traditional meal with healthy dessert” dietary pattern. CI: confidence interval; OR: odds ratio. * *p* < 0.05, ** *p* < 0.01, *** *p* < 0.001.

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
