# Peer review of "Comparative Study of Dietary Patterns by Living Arrangements: The Korea National Health and Nutrition Examination Survey (KNHANES) 2013–2015"

_ijerph, 2020, doi:10.3390/ijerph17072371_

Round 1

Reviewer 1 Report

It is good study and interesting point on living alone. 

Only weak point is big unbalanced between two groups such as living with others and living alone. 15.4:1, the number of living alone group is 401, limitative.

Weather there is some method which can make control on this limitation?

Author Response

Response to Reviewer 1 Comments

Point 1: 

It is good study and interesting point on living alone. Only weak point is big unbalanced between two groups such as living with others and living alone. 15.4:1, the number of living alone group is 401, limitative. Weather there is some method which can make control on this limitation?

Response 1: 

We agree with your opinion. Although weighting somewhat helps overcome the sample size issues for the small number of people living alone, we still felt concerned about the sample size discrepancy.

We planned to match two samples using propensity score matching (PSM) to overcome the wide disparity in the sample sizes of the two groups. However, very limited information about covariates is available to determine a diet pattern using living arrangement status.

Thus, we ultimately decided to conduct this study to identify factors associated with diet pattern by living arrangement. Therefore, we analyzed the influencing factors in each group using a descriptive study to identify the similarities and differences between them. We hope that our study’s findings provide fundamental information that researchers can use to apply PSM using the covariates identified in this study.

We added this thought to the section on limitations as a suggestion for further study.

Location of section: 

Limitations section; Lines 369-373; Page 14

Reviewer 2 Report

Authors report results about interesting study, in which they compared dietary patterns of people living alone with those living with others. This is a secondary data analyses of The Korea National Health and 3 Nutrition Examination Survey 2013–2015 (KNHANES); original data already published. Authors used very large dataset and statistical methods seems sound. Impressively 6,719 Korean adults (18-65y) is included. A major limitation of the study is that a group of focus (people living alone) represent only about 6% of the sample, but their number is still enough for statistical analyses (about 400). The reason for low proportion of people living alone (in comparison to data presented in introduction – other countries) is in another limitation, that KNHANES does not include elderly people, which are more often living alone. Authors fairly mention these limitations in the discussion section. Important observation of the study was newly identified unhealthy dessert dietary pattern in the Korean population, but the data is from 2013-2015. Here are major issues, that would need to be resolved:

  • Data processing and analyses should be described more in detail, to enable study replication. Currently not enough data is provided, that researchers using same dataset would do same analyses again. Specifically, data procedures (section 2.3.3.) are listed only with references, but these methods can have several variations/parameters. Authors should be much more specific and also describe how they handled specifics. Similarly, details in data analyses need to be more detailed. Authors should have in mind that others might wand to use similar approach, and detail practices in data analyses are most welcome in such cases.
  • χ2 statistical analyses is not very strong to handle data provided in Table 4. In the living with others dataset, almost all parameters are significant. Wouldn’t be more reasonable to use more sensitive method, that would highlight those parameters with really strongest effect?
  • The introduction section is oriented into the situation with people lining alone. Authors are mentioning situation in other countries, where a considerable proportion of the population is living alone. Data that about 6% of people in South Korea is living alone give (possibly wrong) impression, that situation in SK is very different. Author explain this only in limitations section, but this should be also mentioned earlier.
  • Is KNHANES the latest dataset, or newer data is also available. Authors should at least explain why this assessment was done with such time delay.

Other issues:

  • There is quite some generalisation in the introduction. I.e. providing exact percentages in lines 41-42 without any further data about data origin is not appropriate.
  • Word ‘validated’ in line 63 should be changed for ‘used’
  • Add reference for statement in L306
  • I think that discussion about ultra-processed foods being a risk for depression is not sufficiently substantiated. Poofs provided in Ref 13 are limited, I suggest that you remove this part of the discussion. There are many risks, which are much more substantiated.
  • Sentence “Eating breakfast is associated with unhealthy dietary patterns regardless of living arrangements.” In L319 is not in a context of the paragraph. Is there a mistake?
  • In discussion (4.12 Practical implications) policy plans are mentioned, starting that these should start in 2019. We are in 2020 now, this paragraph should be updated. Were these plans implicated in practice?

Author Response

Thank you very much for your careful review and another opportunity to submit the revised manuscript. We very much appreciate the thoughtful comments and feel that the paper has been greatly improved.

We have made changes in response to each of the International Journal of Environmental Research and Public Health reviewers’ comments or provided further explanation with evidence. Please see our point-by-point responses in the revision table and in the highlighted text of the revised manuscript. We also received editing services on the revised manuscript. Please see the authors’ response table with this submission.

Round 2

Reviewer 2 Report

Thank you for revising manuscript and providing your feedback. My only remaining issues is in the formulation of discussion section with breakfast frequency. Authors noted that they did not understand the issue raised (L324). I checked revised version and still have same comment:

Your sentence "Eating breakfast was associated with unhealthy dietary patterns." imlied that those eating breakfast are having unhealthy dietary patterns, but latter on it seems that thise NOT eating breakfast are actually having more unhealtha dietary patterns. This can be also seen from results in Table 5, where note eating breakfast (none a week) is statistically difefrent than 5-7 times per week, with Quite high ORs. I wonder if first sentance of the paragraph should be contrary? ( "Not eating breakfast was associated with unhealthy dietary patterns.")
